# Cognitive and neural mechanisms of learning and interventions for improvement across the adult lifespan: A systematic review protocol

Adam John Privitera[1]*, Siew Hiang Sally Ng[1,2], S. H. Annabel Chen[1,3,4,5]

1 Centre for Research and Development in Learning, Nanyang Technological University, Singapore, Singapore, 2 Institute for Pedagogical Innovation, Research, and Excellence, Nanyang Technological University, Singapore, Singapore, 3 School of Social Sciences, Nanyang Technological University, Singapore, Singapore, 4 National Institute of Education, Nanyang Technological University, Singapore, Singapore, 5 Lee Kong Chian School of Medicine, Nanyang Technological University, Singapore, Singapore

* aprivite@connect.hku.hk

## Abstract

### Background

There continues to be growing interest in the Science of Learning including identifying applications for findings from this work outside the laboratory to support learning. Presently, there exists a gap in our understanding of learning during healthy adulthood as well as effective ways in which that learning can be improved. Developing a more comprehensive understanding of learning during adulthood, and effective ways of improving that learning, are crucial goals given the impact of a rapidly aging global population. The main objective of the proposed systematic review is to identify and synthesize all recent cognitive and brain research investigating learning across the adult lifespan.

### Methods

Searches will be performed across Scopus, Web of Science, and ProQuest databases. Both published and unpublished literature will be screened for inclusion. Included articles will be limited to research in healthy adult samples reporting measures of learning-related cognition, brain structure or function and their relationship with age, or the impact of interventions to improve learning. All steps of the review will be performed by three trained reviewers. Tabular, narrative, and quantitative syntheses will be provided based on the characteristics of included studies.

### Discussion

Findings from the proposed review will contribute to our understanding of learning in adulthood. Additionally, this review will identify research gaps in need of further investigation and relevant findings for translation, informing the scope of future funding priorities in the Science of Learning.

**Data Availability Statement:** No results are reported in the current protocol manuscript. This

protocol was preregistered with the Center for Open Science (https://osf.io/d9yev/).

**Funding:** This project is funded by the Singapore Ministry of Education (#MOE-SOL-2023T001). The funders had no role in study design, data collection and analysis, decision to publish, or preparation of the manuscript.

**Competing interests:** The authors have declared that no competing interests exist.

# Introduction

## Rationale

Researchers, practitioners, and the general public have continued to exhibit growing interest in the Science of Learning, "the scientific study of the underlying bases of learning with the goal of describing, understanding, or improving learning across developmental stages and diverse contexts" [1]. Recent advances in the cognitive and neural sciences have illuminated our understanding of the principles, processes, and mechanisms of learning across a diverse range of species, including our own [e.g., 2]. Findings from these studies support the conclusion that learning is a complex lifelong process involving the interplay of cognitive, neural, and other factors [e.g., 3]. Furthermore, the experience of learning modifies underlying cognitive processes and brain structure and function in a manner that impacts on future cognition and behavior [4]. Given the relevance of this work outside the research laboratory, there has been growing interest in the application of findings from the Science of Learning to a number of other fields, including education [e.g., 5–7].

While there is tremendous interest in understanding human learning, previous investigations have generally not been developmentally comprehensive in their focus. Existing studies on the Science of Learning have predominantly investigated samples drawn from developmental stages during early and late life [8], although some notable exceptions do exist [e.g., 9]. This same pattern can be observed across published reviews with most focusing on research in samples of younger children [e.g., 10], or older adults, including clinical populations diagnosed with neurodegenerative diseases [11]. Collectively, this has created a gap in our understanding of learning during healthy adulthood as well as effective ways in which that learning can be improved. Given that the human brain retains the capacity for plasticity across the adult lifespan [12], and that adult learning trajectories can be altered based on the influence of both internal and external factors [13], there is significant value in better understanding learning during this period of development in the interests of maximizing human potential.

Developing a more comprehensive understanding of learning during adulthood, and effective ways of improving that learning, are crucial goals given current global demographic trends. Many nations around the world are continuing to struggle in addressing the significant consequences of a rapidly aging population [14]. This issue is further complicated by the dynamic nature of the modern workforce where regular career changes are becoming the norm [15]. Consequently, governments have recently shifted attention and resources toward extending and translating findings from the Science of Learning in order to inform educational policy and intervention strategies. One of the overarching goals of this shift is to support the development of a citizenry that is both productive and agile into late adulthood. Presently, the absence of a comprehensive understanding of learning across adulthood in healthy populations places a significant hurdle on the path to accomplishing this goal. The conduct of an exhaustive review synthesizing existing research on the cognitive and neural mechanisms of learning and interventions to improve learning in healthy adults would contribute significantly to the accomplishment of this goal, with the additional benefit of identifying areas in need of further research.

## Objectives

The main objective of the proposed systematic review is to identify and synthesize all recent cognitive and brain research investigating learning across the adult lifespan. This review will provide an overview of the principles, processes, and mechanisms of learning in healthy adults. Additionally, this review aims to synthesize all recent findings on interventions to improve

learning in adults. Findings from the proposed review will have implications for our understanding of learning in adulthood, the scoping of funding calls for future Science of Learning research, and for public education and outreach efforts to support lifelong learning.

## Research questions

The primary research question (RQ) of the proposed review is, "How does learning change across a healthy adult's lifespan?" The research sub-questions are:

1. What learning-related cognitive changes occur in healthy adults as they age?

    a. Which specific dimensions of cognitive function have been investigated?

    b. How do changes in these dimensions influence learning?

    c. What factors influence these changes?

2. What learning-related brain changes occur in healthy adults as they age?

    a. Which specific dimensions of brain structure/function have been investigated?

    b. How do changes in these dimensions influence learning?

    c. What factors influence these changes?

3. What interventions based on evidence from 1 and 2 are most effective in the improvement of learning in healthy adults?

    a. What are the reported effects of interventions on cognitive function?

    b. What are the reported effects of interventions on brain structure/function?

    c. What are the mediating and moderating factors that influence the efficacy of interventions?

4. What are the current knowledge gaps identified in 1–3?

For the purpose of the present review, learning will be broadly defined to include traditional measures related to change in achievement or attainment, as well as changes in underlying cognitive and neural processes such as executive function, intelligence, and brain network activation.

## Methods

### Registration

The protocol for the proposed review was designed following recommendations provided in the Preferred Reporting in Systematic Reviews and Meta-Analysis Protocols (PRISMA-P) checklist [16] and was registered with the Centre for Open Science (https://osf.io/d9yev/). Deviations from the registered protocol will be explicitly noted in the final version of the paper. A timeline for the conduct of the proposed review is presented in Table 1.

### Eligibility criteria

Studies will be included if they meet the following criteria:

• Journal articles, theses, dissertations, conference papers, and preprints

• Cross-sectional, longitudinal, and sequential studies or research syntheses

**Table 1. Timeline for the proposed review.**

| Phase | Task | Dates | Status |
|---|---|---|---|
| 1 | Form research questions | SEP 2023 | Completed |
| 2 | Protocol registration | SEP 2023 | Completed |
| 3 | Search strategy piloting | SEP-OCT 2023 | Completed |
| 4 | Selection and screening | NOV 2023-JAN 2024 | Completed |
| 5 | Extraction | FEB-MAR 2024 | In Progress |
| 6 | Synthesis and analysis | MAR-MAY 2024 | Not Started |
| 7 | Writing the review | FEB-JUN 2024 | Not Started |

- Published or posted between January 1st, 2012 and September 1st, 2023

- Reports one or more of the following:

  ○ RQ1: Relationship between measure(s) of learning-related cognition and age.

  ○ RQ2: Relationship between measure(s) of learning-related brain structure or function and age.

  ○ RQ3: Impact of any intervention aimed at improving learning.

- Focused on healthy adult humans

- Reports findings (quantitative or qualitative) from participants age $\geq$ 20 years either exclusively or as a separate analysis from a wider age group.

- Written in English

- Full text available

Studies will be excluded if any of the above criteria are not met or if the document is a purely theoretical paper, non-systematic narrative review, magazine article, abstract, protocol, preregistration, blog post, editorial, commentary, corrigendum, book, or book review.

## Information sources

The search for relevant literature will occur using Scopus, Web of Science (Core Collection and Preprints), and ProQuest databases. These databases were selected due to their extensive indexing of relevant published work as well as conference papers, theses, dissertations, and preprints. Search dates will be limited from January 1st, 2012 to September 1st, 2023. Studies identified across all databases will be pooled and filtered for duplicate results prior to screening for inclusion. If the full text for an article cannot be located, corresponding authors will be contacted. If no reply is received after two weeks, the study in question will be excluded.

## Search strategy

For each database used, search strings will be written following the database-specific requirements related to Boolean operators and search limiters. Strings for each database will contain a combination of terms aimed at identifying all potentially relevant articles including "learning" and "adult", as well terms related to different measures of interest including "memory", "intelligence", "brain", and "executive function". Finally, additional terms will be included using the "NOT" operator to prevent the inclusion of large numbers of irrelevant articles including "dementia", "impairment", and "brain damage". When appropriate, "wildcard" search terms will be included to allow for inclusion of highly similar terms (e.g., "adult*" for "adult",

**Table 2. Example draft web of science (core collection) search strategy.**

| Aim | Field(s) | Operator | Search term(s) |
|---|---|---|---|
| To identify all sources investigating learning | Title OR abstract | N/A | learn* OR education* |
| To limit results to only those investigating the cognitive or neural aspects of learning | Title OR abstract | AND | cogniti* OR neur* OR brain OR plasticity OR memory OR intelligence OR "executive function*" OR thinking OR motivation |
| To further limit results to those conducted in samples of adults | Title OR abstract | AND | adult* OR senior* OR "middle age" OR "middle aged" OR elder* OR lifespan OR "life span" OR lifelong OR "life long" |
| To further limit results based on commonly used terms that are likely to indicate an irrelevant paper if used in the abstract or title | Title OR abstract | NOT | ("artificial intelligence" OR "machine learning" OR "computer science*" OR "learning algorithm*" OR prison* OR disorder* OR disab* OR aphasia OR addiction OR impair* OR illness* OR injury OR trauma* OR patholog* OR autis* OR stroke OR ischemi* OR damage* OR lesion OR cancer* OR suicide OR depress* OR epilep* OR diabet* OR rat OR rats OR mice OR mouse OR rodent* OR monkey* OR drosophila OR bird* OR fish*OR neonat* OR infant* OR infancy |
| To further limit results based on commonly used terms that are likely to indicate an irrelevant paper if used only in the title | Title | NOT | child* OR teenage* OR animal* OR adolescen* OR dementia OR alzheimer* OR MCI OR disease* |
| To limit results to articles, conference proceedings, or review papers | N/A | AND | NOT (DT = = ("MEETING ABSTRACT" OR "BOOK CHAPTER" OR "EDITORIAL MATERIAL" OR "CORRECTION" OR "BOOK REVIEW" OR "LETTER" OR "DATA PAPER" OR "NEWS ITEM" OR "BOOK" OR "RETRACTED PUBLICATION" OR "RETRACTION" OR "THEATER REVIEW")) |
| To limit results to only those published in English | N/A | AND | LA = = "ENGLISH" |

\* Indicates truncation of a search term.

"adulthood", "adults"). An example draft search strategy for Web of Science (Core Collection) is presented in Table 2. Searches will be limited to title and abstract fields and will be run using the "exact search" option or equivalent for each database. Search string performance will be evaluated based on the recovery of known sentinel papers [9,17,18].

## Study records

**Data management.** Search records for each database will be exported directly as Research Information Systems (RIS) formatted files. In the event multiple files are exported from a single database due to export limits, citations will be combined into a single RIS file for each database. Files will be imported into Covidence. Covidence was selected due to a range of features that streamline the review process including deduplication of uploaded citations, collaborative screening, selection, and extraction, and automatic generation of PRISMA flowcharts [19]. We also selected Covidence because of its security features. Covidence employs encryption techniques to secure data transmission and storage, and ensures secure user authentication to prevent unauthorized access. In addition, Covidence implements role-based access control which allows the principal investigator to assign different levels of access and permissions to users based on their role within the review team. The cloud-based platform maintains regular data backups to mitigate data loss, adhering to relevant data protection regulations such as the General Data Protection Regulation (GDPR) to provide user privacy and ensure compliance with legal requirements.

**Selection process.** Selection of studies will be performed by three trained reviewers. Full details of the screening process will be summarized in PRISMA flow diagram format [16]. Screening will initially occur at the title and abstract level for articles. Next, full text versions of all candidate articles will be retrieved from the Internet, institutional libraries, or

corresponding authors. Retrieved studies will then undergo full text screening, applying our inclusion and exclusion criteria. During title/abstract and full text screening, agreement between two independent reviewers will be required in order for a document to be included in the review. If two reviewers disagree about the inclusion of a document, the third reviewer will settle the conflict. In the event a preprint was later published as a journal article, only the journal article will be included. Search and selection of relevant articles is expected to be completed by January 31st, 2024.

**Data collection process.** Extraction of relevant data will be guided by a customized spreadsheet template in Covidence. If relevant data are not reported in an included study, corresponding authors will be contacted. If no reply is received after two weeks, all attempts will be made to include the study in analyses. Extraction will be completed by three trained reviewers. For all included studies, consensus between two independent reviewers will be assessed through comparison of extracted data. If two reviewers disagree about any extracted items, the third reviewer will settle the disagreement.

## Data items

The following data will be extracted from all selected studies: 1) publication year; 2) authors; 3) source format (e.g., journal article); 4) country the study was conducted in; 5) study setting; 6) study design; 7) study duration (if applicable); 8) population of interest 9) number of participants by sex per group; 10) average age of participants per group; 11) sociodemographic background of participants per group; 12) cognitive, brain, or learning outcome(s) reported; 13) additional variables, and; 14) main results. Additionally, 15) intervention format; 16) comparison group, 17) outcome of interest, and; 18) measure of impact of intervention on outcome of interest will be extracted from intervention studies. In the event necessary data were not reported in the original articles, corresponding authors will be contacted. Authors not responding to a request for data within two weeks will not be contacted again and the study in question will be included with incomplete details. Extracted data will be summarized in tabular format organized by research question. Data from single studies reporting findings relevant to multiple research questions (e.g., study that reports the relationship between both cognitive and brain measures with learning), will be summarized under a "combined studies" heading in the synthesis table with relevant findings contributing to their respective narrative synthesis or, in the case of intervention studies, quantitative synthesis. Extraction table finalization is expected to be completed by March 31st, 2024.

## Outcomes and prioritization

The primary outcomes of the proposed review are measures of the relationship between cognition or brain structure/function and age (RQ1 & RQ2), and learning difference scores or qualitative descriptions from comparisons between control and experimental groups or pre- and post-intervention timepoints (RQ3). Cognitive measures can include reaction time or accuracy rates from behavioral tasks (e.g., Simon task, n-back task), scores from cognitive assessments (e.g., intelligence test score), or self-reported measures (e.g., Behavior Rating Inventory of Executive Function for Adults). Brain measures can include data from any neuroanatomical or neurophysiological instrument related to structure (e.g., magnetic resonance imaging [MRI]) or function (e.g., electroencephalogram [EEG]). Measures of age can include those that are quantified (e.g., age in years) as well as categorical labels (e.g., young adults and elders). Measures of learning can either be objectively measured (e.g., performance on a memory task, motor skill performance) or self-reported (e.g., GPA, educational attainment). For RQ3, the

impact of an intervention can be represented as either a quantitative measurement or qualitative assessment of change in reported learning-related outcome.

## Risk of bias in individual studies

We will evaluate the risk of bias in each individual study by employing the relevant checklists provided by the Joanna Briggs Institute [20]. Every included study will receive a rating of "Yes," "No," "Unclear," or "NA" for each criterion assessed. A team of three trained reviewers will conduct this assessment, and inter-reviewer reliability will be calculated across 20% of the included studies. No studies will be excluded from the planned review based on the results of the risk of bias assessment.

## Data synthesis

Regarding RQ3, if there are at least two studies that report the impact of a specific intervention on learning, we will conduct a quantitative synthesis. Analyses will be performed in jamovi [21]. Each type of intervention will undergo separate analysis. Either Hedge's *g* or Fisher's *r*-to-*z* transformed correlation coefficient will be used as the outcome measure depending on the outcome reported in a given study. To model the data, we will utilize a random-effects model and will estimate heterogeneity (i.e., $tau^2$) using the restricted maximum-likelihood estimator [22]. Additionally, heterogeneity will be reported as the $I^2$ statistic [23]. We will also assess the presence of outliers or influential data points using studentized residuals and Cook's distances. For each model, we will generate forest plots, which will include the pooled effect size. However, if the data reported across the included intervention studies exhibit significant heterogeneity that prevents a quantitative synthesis, we will assess the effectiveness of interventions using effect direction analysis [24,25]. Finally, if we do not identify a sufficient number of studies for a particular intervention or outcome (i.e., fewer than 2), we will provide a narrative synthesis.

## Meta-biases

If data from included studies supports the conduct of a quantitative synthesis for RQ3, publication bias will be assessed through generation of Egger's statistics for each intervention type [26], and funnel plot visual analysis. Both Egger's statistics and funnel plots will be reported in the final version of the proposed review.

## Confidence in cumulative evidence

For RQ3, we will evaluate the quality of evidence following the methodology outlined by the Grading of Recommendations Assessment, Development and Evaluation (GRADE) working group [27]. This assessment will encompass several domains, including risk of bias, consistency, precision, and publication bias. The overall quality of evidence will be categorized using the GRADE certainty ratings [28,29], which include:

- *High*: indicating a high level of confidence that the true effect aligns closely with the estimated effect.

- *Moderate*: signifying a belief that the true effect is likely to be close to the estimated effect.

- *Low*: suggesting that the true effect might significantly differ from the estimated effect.

- *Very low*: indicating a high likelihood that the true effect significantly differs from the estimated effect.

Quality of evidence assessment will be modified accordingly in the event that we are unable to perform a quantitative synthesis.

## Limitations

The proposed systematic review's contributions are potentially limited by the exclusion of resources written in languages other than English. As a result, developments in the Science of Learning from parts of the world where English proficiency is lower will likely be missed. Additionally, because the proposed review will only focus on research in healthy adult samples, findings are likely not generalizable to adults living with psychiatric or neurological conditions that may impact on learning. Finally, given the diverse number of ways in which learning can be operationalized, heterogeneity across studies may prevent strong, broadly-focused conclusions from being drawn regarding changes in learning across the healthy adult lifespan.

## Conclusion

Findings from the proposed systematic review will contribute to our developing understanding of the complex nature of learning in healthy adult populations. Synthesizing an overview of key findings from this area of investigation will effectively map out what is currently known, supporting policy makers and funding agencies in the identification of research gaps that could be bridged by further use-inspired basic and applied research. Identification of these gaps can inform the scope of future funding priorities in the Science of Learning. Finally, this work will aid in the identification of relevant findings for translation outside of the research laboratory, as well as efficacious programs ready for development and scaling.

## Supporting information

**S1 File. PRISMA checklist.** Completed PRISMA-P (Preferred Reporting Items for Systematic review and Meta-Analysis Protocols) checklist.
(PDF)

## Author Contributions

**Conceptualization:** Adam John Privitera, Siew Hiang Sally Ng, S. H. Annabel Chen.

**Funding acquisition:** Adam John Privitera, Siew Hiang Sally Ng, S. H. Annabel Chen.

**Methodology:** Adam John Privitera.

**Supervision:** Adam John Privitera.

**Writing – original draft:** Adam John Privitera.

**Writing – review & editing:** Adam John Privitera, Siew Hiang Sally Ng, S. H. Annabel Chen.

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
