## [Decision Letter · Decision Letter 0]

18 Mar 2024

PONE-D-23-40448Cognitive and Neural Mechanisms of Learning and Interventions for Improvement Across the Adult Lifespan: A Systematic Review ProtocolPLOS ONE

Dear Dr. Privitera,

Thank you for submitting your manuscript to PLOS ONE. After careful consideration, we feel that it has merit but does not fully meet PLOS ONE’s publication criteria as it currently stands. Therefore, we invite you to submit a revised version of the manuscript that addresses the points raised during the review process.

We look forward to receiving your revised manuscript.

Kind regards,

Muhammad Shahzad Aslam, Ph.D.,M.Phil., Pharm-D

Academic Editor

PLOS ONE

 [This project is funded by the Singapore Ministry of Education (#MOE-SOL-2023T001)].  

Please respond by return e-mail so that we can amend your financial disclosure and competing interests on your behalf.

Additional Editor Comments:

 the question regarding intervention lacks concrete sub-questions compared to the detailed queries of RQ1 & 2. Suggesting a detailed question to investigate the precise cognitive and brain alterations induced by interventions could enhance the linkage to identified learning-related mechanisms, offering clearer guidance for the study's intervention part.

Reviewers' comments:

Reviewer's Responses to Questions

**Comments to the Author**

1. Does the manuscript provide a valid rationale for the proposed study, with clearly identified and justified research questions?

Reviewer #1: Yes

2. Is the protocol technically sound and planned in a manner that will lead to a meaningful outcome and allow testing the stated hypotheses?

Reviewer #1: Yes

3. Is the methodology feasible and described in sufficient detail to allow the work to be replicable?

Reviewer #1: Yes

4. Have the authors described where all data underlying the findings will be made available when the study is complete?

Reviewer #1: Yes

5. Is the manuscript presented in an intelligible fashion and written in standard English?

Reviewer #1: Yes

6. Review Comments to the Author

You may also provide optional suggestions and comments to authors that they might find helpful in planning their study.

Reviewer #1: The manuscript meticulously presents a protocol for a systematic review, pinpointing a significant knowledge gap in adult learning. It adeptly outlines the study's importance and proposes well-structured questions on learning-related changes, focusing on cognitive and brain changes. However, the question regarding intervention lacks concrete sub-questions compared to the detailed queries of RQ1 & 2. Suggesting a detailed question to investigate the precise cognitive and brain alterations induced by interventions could enhance the linkage to identified learning-related mechanisms, offering clearer guidance for the study's intervention part.

7. PLOS authors have the option to publish the peer review history of their article (what does this mean?). If published, this will include your full peer review and any attached files.

Reviewer #1: No

---

## [Author Response · Author response to Decision Letter 0]

19 Mar 2024

Response to Reviewers

Paper: Cognitive and Neural Mechanisms of Learning and Interventions for Improvement Across the Adult Lifespan: A Systematic Review Protocol

Journal: PLOS ONE

General Author Comments

C1: Please update funding statement

R1: We apologize for not proving enough detail. The fundings statement should read as following: 

This project is funded by the Singapore Ministry of Education (#MOE-SOL-2023T001). The funders had no role in study design, data collection and analysis, decision to publish, or preparation of the manuscript.

Editor/Reviewer 1 (Comments were identical)

C1: Question regarding intervention lacks concrete sub-questions compared to the detailed queries of RQ1 & 2. Suggesting a detailed question to investigate the precise cognitive and brain alterations induced by interventions could enhance the linkage to identified learning-related mechanisms, offering clearer guidance for the study's intervention part.

R1: Thank you for your comment. We have provided specific sub-questions for RQ3.

---

## [Editor Report · Decision Letter 1]

25 Mar 2024

Cognitive and Neural Mechanisms of Learning and Interventions for Improvement Across the Adult Lifespan: A Systematic Review Protocol

PONE-D-23-40448R1

Dear Dr. Privitera,

We’re pleased to inform you that your manuscript has been judged scientifically suitable for publication and will be formally accepted for publication once it meets all outstanding technical requirements.

Kind regards,

Muhammad Shahzad Aslam, Ph.D.,M.Phil., Pharm-D

Academic Editor

PLOS ONE
---

## [Editor Report · Acceptance letter]

26 Apr 2024

PONE-D-23-40448R1 

PLOS ONE

Dear Dr. Privitera, 

I'm pleased to inform you that your manuscript has been deemed suitable for publication in PLOS ONE. Congratulations! Your manuscript is now being handed over to our production team.

Kind regards, 

on behalf of

Dr. Muhammad Shahzad Aslam 

Academic Editor

PLOS ONE